# The Moderating Role of Personality on the Effects of Concentration-, Ethics- and Wisdom-Based Meditation Practices for Well-Being and Prosociality

**DOI:** 10.3390/healthcare13233044

**Published:** 2025-11-25

**Authors:** Matthew Furnell, William Van Gordon, James Elander

**Affiliations:** School of Psychology, University of Derby, Derby DE22 1GB, UK

**Keywords:** mindfulness-based intervention, personality traits, big five, Buddhist meditation practices, Buddhism, prosocial behaviour

## Abstract

**Highlights:**

**What are the main findings?**
Individuals high in neuroticism showed greater prosocial gains when mindfulness interventions included both ethics- and wisdom-based meditation practices.

**What are the implications of the main findings?**
Incorporating wisdom-based practices alongside traditional ethics-based components may enhance the effectiveness of mindfulness-based interventions for certain personality profiles.Tailoring mindfulness interventions according to personality traits may improve their impact in both clinical and non-clinical contexts.

**Abstract:**

**Objectives:** The majority of mindfulness research has focused on group-level effects, with limited attention to how engagement with specific meditation practices interacts with individual differences such as personality. This study examined whether Big Five traits moderated the effects of two mindfulness-based interventions (MBIs) on well-being and prosocial outcomes: one incorporating concentration- and ethics-based practices (MBI-CE) and another incorporating concentration-, ethics- and wisdom-based practices (MBI-CEW). **Methods:** Fifty-six participants were randomly assigned to MBI-CE (*n* = 21), MBI-CEW (*n* = 18), or a waitlist Control group (*n* = 17). Both interventions followed an 8-week programme. Pre- and post-intervention assessments measured changes in well-being and prosocial outcomes. Baseline personality traits were analysed as continuous variables using linear regression to test moderation effects, with ANCOVA sensitivity analyses conducted to assess robustness. **Results:** Participants high in neuroticism showed significantly greater prosocial gains under MBI-CEW compared to MBI-CE and Control (MBI-CEW × Neuroticism: β = 2.09, *p* = 0.021). High agreeableness moderated well-being outcomes for both interventions (MBI-CE: β = 1.873, *p* = 0.044; MBI-CEW: β = 2.701, *p* = 0.012), while high openness moderated well-being only for MBI-CEW (β = 2.478, *p* = 0.045). These findings suggest that the additional incorporation of wisdom-based practices (such as contemplations on interdependence and the emptiness of self), in combination with ethics-based practices, may enhance the prosocial effects of MBIs for individuals high in neuroticism, who are prone to interpersonal sensitivity and self-protective tendencies. **Conclusions:** Findings underscore the importance of moving beyond one-size-fits-all approaches to mindfulness. Tailoring MBIs to personality profiles, and integrating wisdom-based elements, in addition to ethics-based practices, may support more effective and sustainable outcomes in both clinical and non-clinical contexts.

## 1. Introduction

There has been a growing interest in understanding the relationship between mindfulness and personality, given their shared implications for psychological well-being and behaviour [1]. The majority of this research has focused on dispositional mindfulness, defined as the innate capacity to maintain attention to present-moment experiences with an open and nonjudgmental attitude [2], and its association with the five-factor (Big Five) model of personality, comprising traits of neuroticism, extraversion, openness to experience, agreeableness, and conscientiousness [3].

Although the relationship between dispositional mindfulness and personality traits has been extensively explored [4,5,6], much less attention has been given to understanding whether, and in what ways, the active practice of mindfulness meditation is related to personality traits [7]. It is important to distinguish between dispositional mindfulness and mindfulness meditation, as these constructs, while potentially overlapping, are not synonymous [8]. More specifically, dispositional mindfulness refers to a measurable, semi-constant trait that reflects an individual’s capacity to remain present and mindful during everyday experiences (e.g., as assessed by the Five Facet Mindfulness Questionnaire [9] and the Mindful Attention Awareness Scale [2]), while mindfulness meditation commonly refers to the specific meditation practices and techniques actively employed during mindfulness-based interventions (MBIs; ref. [10]). Understanding how personality traits may moderate the effects of meditation practices within MBIs is particularly important, given the key role personality traits play in influencing the outcomes of psychological interventions [11,12].

Specific personality traits are associated with various psychological outcomes, suggesting that certain traits may either enhance or hinder the effectiveness of psychological interventions [13]. For example, neuroticism, along with related factors such as intolerance to uncertainty, has been shown to influence how individuals respond to interventions, highlighting the need to account for these traits in the design of intervention programmes [14]. Although a growing body of research demonstrates that MBIs can yield significant benefits for psychological well-being and behaviour, however, these effects are often generalised at the population level, assuming homogeneous effects of training for every individual based on group-averaged results [1]. Some studies have examined individual differences in mindfulness-related outcomes by gender [15]; however, the influence of personality traits on individual variability in MBI outcomes has not yet been systematically investigated [1].

Every individual possesses a unique combination of personality traits and needs, making it important to adapt MBIs to specific contexts and personal characteristics [12]. From a practical standpoint, recognising individual differences in responsiveness to mindfulness training is vital for implementing MBIs in clinical and public settings, as it helps identify which subgroups are most or least likely to benefit from the intervention [1]. Although research on the moderating effects of the Big Five personality traits on MBI outcomes remains limited, there have been some studies investigating this relationship. For instance, Krick and Felfe [16] examined whether the effectiveness of an MBI for police officers varied based on their personality traits, finding that individuals with higher levels of neuroticism and openness experienced greater benefits. Similarly, de Vibe et al. [17] examined how personality traits moderated the effects of MBIs on students, reporting that those with higher neuroticism benefitted more from the intervention, showing lower levels of mental distress and increased well-being, while those with higher conscientiousness experienced a stronger effect of the intervention on study stress. Additionally, Winning and Boag [18] found that specific personality traits, such as conscientiousness and extraversion, moderated the effects of an MBI on cognitive empathy.

Although most MBIs follow a standardised 8-week duration as originally employed in Kabat-Zinn’s mindfulness-based stress reduction (MBSR) programme [19], the specific practices and approaches within MBIs can differ significantly [10]. For example, first-generation MBIs (i.e., MBIs that primarily incorporate concentration-based meditation practices; MBI-C) emphasise present-centred awareness, while second-generation MBIs additionally incorporate ethics-based and/or wisdom-based practices as core components of the intervention [20]. Concentration-based practices include meditation techniques such as awarenesses of breath and body scans, ethics-based practices include techniques such as loving-kindness and compassion meditation, and wisdom-based practices involve meditations on concepts such as impermanence, interdependence, and emptiness [21].

Previous research has suggested that personality traits can predict an individual’s preference for specific meditation practices [22]. For instance, individuals with higher levels of empathy are more likely to prefer loving-kindness meditation (an ethics-based practice), while those with greater non-reactivity and non-judgement of present-moment experiences tend to favour open monitoring techniques (a concentration-based practice) [22]. These findings indicate that people are drawn to meditation practices that align with their personality traits, as these traits reflect capacities that are particularly suited to the demands of specific techniques. Previous studies have also examined which personality traits are most associated with a willingness to engage in various meditation practices, such as yoga, sitting meditation, informal meditation, and body scanning [23]. Findings revealed that the trait of openness was most strongly linked to engagement with meditation techniques both during and after the MBI, while agreeableness was strongly related to engagement with meditation techniques during the intervention [23].

Such studies have primarily focused on the preferences associated with different personality traits, without examining how these traits might influence the outcomes of specific meditation techniques. To the best of the present researchers’ knowledge, no study has explicitly investigated personality traits as moderators for the effectiveness of different meditation practices, though some studies have hinted at this possibility. For instance, Chen and Jordan [24] found that an MBI incorporating only concentration-based meditation practices (MBI-C) led individuals with low trait empathy to donate significantly less money than those in a control group. Notably, this unintended reduction in charitable behaviour was mitigated for individuals of an MBI that included concentration-based practices, ethics-based practices (e.g., loving-kindness meditation) and wisdom-based practices (e.g., contemplating interdependence). However, as this intervention incorporated both ethics- and wisdom-based practices, it remains unclear whether the observed mitigation of reduced charitable behaviour was driven solely by the inclusion of ethics-based practices or by the combined interplay of ethics- and wisdom-based techniques.

These findings align with concerns raised by some scholars that MBIs focused solely on concentration-based practices may inadvertently promote self-indulgence and selfish individualism (i.e., an excessive focus on oneself at the expense of others), potentially reducing prosocial behaviour and wholesome actions in certain individuals [8,25]. This underscores the need for further research to identify which individuals may be most vulnerable to these unintended effects and to explore whether the inclusion of ethics-based practices, or a combination of ethics- and wisdom-based practices, can effectively mitigate these negative outcomes.

Therefore, this study adopts a hypothesis-driven approach to investigate whether the Big Five personality traits moderate the effects of two distinct MBIs on psychological well-being and prosocial behaviour. One intervention incorporated concentration- and ethics-based practices (MBI-CE), while the other included concentration-, ethics-, and wisdom-based practices (MBI-CEW). The primary aim was to assess whether the inclusion of ethics-based practices, or the combination of ethics- and wisdom-based practices, could enhance outcomes for individuals with different personality traits, while also exploring whether these practices could mitigate any unintended negative effects.

### 1.1. Study Hypotheses

#### 1.1.1. Openness to Experiences

Openness to experience reflects curiosity, cognitive and emotional receptivity, and a willingness to engage with novel and diverse perspectives and practices [1]. Individuals high in openness tend to be imaginative, broad-minded, and curious [3], and they are often more attuned to inner feelings and emotions [5]. This trait is particularly relevant for meditation engagement, as individuals with high openness are more likely to try, persist with, and explore a variety of contemplative techniques during and after an MBI [23]. Concerns have been raised about whether non-Buddhist and secular individuals can effectively understand, access, and embrace Buddhist-based practices [26]. High openness to experience may help individuals overcome these potential barriers by fostering curiosity and receptivity to unfamiliar concepts. While the inclusion of Buddhist ethics-based practices is unlikely to significantly impact engagement, given that principles such as “no harm” and compassion are universal and found across many global religions and cultural traditions [27], Buddhist wisdom-based practices may present greater challenges. These practices, which emphasise concepts like interdependence and non-self, are likely to be less familiar to a secular or non-Buddhist audience, potentially requiring higher levels of openness to fully engage with and comprehend them.

Thus, we hypothesise (Hypothesis 1) that individuals high in openness to experience will benefit more from the MBI-CEW intervention compared to those with lower openness in the same group, as well as compared to individuals in the MBI-CE group, across both well-being and prosocial behaviour outcomes.

#### 1.1.2. Conscientiousness Hypothesis

Conscientiousness reflects a general tendency toward self-regulation, goal-directed behaviour, and adherence to social norms related to impulse control [28]. Individuals high in conscientiousness are typically dependable, responsible, self-disciplined, and achievement-oriented [3]. Conscientiousness has been linked to motivation to learn, a process variable associated with intervention effectiveness [29], suggesting that conscientiousness may facilitate engagement with and adherence to contemplative training. Although one study reported no association between conscientiousness and formal meditation practice time within an MBI [23], other work has found that higher conscientiousness predicts greater psychological improvements following mindfulness interventions [17], consistent with a potential moderating role. In the present context, both interventions (MBI-CE and MBI-CEW) require sustained effort and practice fidelity; thus, conscientiousness should broadly support engagement and benefit across well-being and behaviour.

Thus, we hypothesise (Hypothesis 2) that individuals high in conscientiousness will experience greater benefits from both interventions (MBI-CE and MBI-CEW), with no significant differences anticipated between the two interventions, across both well-being and prosocial behaviour outcomes.

#### 1.1.3. Extraversion Hypothesis

Extraversion is characterised by sociability, talkativeness, and assertiveness [3] and is often linked to greater experience of positive emotions [30]. Extraverted individuals are commonly described as bold, energetic, and enthusiastic [31]. Extraverts often prefer interventions with interpersonal engagement [1] and may find slow-paced, introspective practices less inherently appealing [5]. Ethics-based practices such and loving-kindness and compassion meditation are outward-facing techniques focused on developing interpersonal connection with others at an emotional level, whereas wisdom-based practices are much more interoceptive, cultivating awareness associated with non-dual states of being [32]. Given that wisdom-based practices require a greater focus on introspection, they may not be suited to individuals with high extraversion, who are arguably less comfortable with inward-focused activities.

Thus, we hypothesise (Hypothesis 3) that individuals high in extraversion will benefit less from the MBI-CEW intervention compared to those with lower extraversion in the same group, as well as compared to individuals in the MBI-CE group, across both well-being and prosocial behaviour outcomes.

#### 1.1.4. Agreeableness Hypothesis

Agreeableness encompasses being cooperative, supportive, caring, good-natured, and concerned for others [3]. Agreeable individuals are pleasant to interact with and tend to maintain harmonious relationships [33]. Previous research has shown in an 8-week MBI, higher agreeableness predicted more frequent practice during the programme, though not after it ended [23], suggesting greater willingness to follow instructors’ guidance for daily meditation practice during meditation interventions [1]. Both MBI-CE and MBI-CEW include multiple contemplative practices across weekly sessions, and agreeable individuals may be especially receptive to this variety and compliant with practice recommendations. This is particularly relevant as prior work links greater engagement in meditative practice to improved training effects [34,35].

Thus, we hypothesise (Hypothesis 4) that individuals high in agreeableness will experience greater benefits from both interventions (MBI-CE and MBI-CEW), with no significant differences anticipated between the two interventions.

#### 1.1.5. Neuroticism Hypothesis

Neuroticism refers to a general tendency to experience negative emotions and respond poorly to stressors [36]. Individuals high in neuroticism are often characterised by anxiety, self-consciousness, moodiness, and insecurity [3], making them more prone to psychological distress and difficulties coping with stress [37]. Meditation practices can support psychological well-being in neurotic individuals by training them to recognise the arising and passing of difficult emotions, fostering less impulsive and more adaptive responses [4]. Consequently, studies have found that highly neurotic individuals often benefit most significantly from MBIs, as they typically have greater room for improvement in managing anxiety and stress, as well as improvements in subjective well-being [17,38].

However, neuroticism is also strongly correlated with vulnerable narcissism [39], and some contemplative techniques, when practiced without an ethical or wisdom-based component, may risk reinforcing vulnerable narcissistic traits, such as self-protectiveness and social withdrawal [18]. In other words, MBIs that focus solely on concentration-based meditation practices may reduce stress reactivity for highly neurotic individuals but could inadvertently diminish prosocial behaviour by heightening self-focused processing without sufficient context. Wisdom-based practices, on the other hand, have been proposed as particularly effective in mitigating these risks [10]. By cultivating decentring and perspective-taking, wisdom-based practices can reduce maladaptive self-focus, thereby weakening defensive self-protectiveness and rumination, which are hallmarks of vulnerable narcissism [40].

Thus, we hypothesise (Hypothesis 5) that individuals high in neuroticism will benefit more from both the MBI-CE and MBI-CEW interventions compared to those with lower neuroticism in the same groups, specifically in terms of well-being outcomes. However, for prosocial outcomes, only individuals high in neuroticism in the MBI-CEW intervention are expected to show significant increases in prosocial outcomes.

## 2. Methods

### 2.1. Participants

Following ethical approval, a total of 67 participants enrolled in the study, with 56 retained for the final analysis after exclusions for incomplete personality data. The final sample comprised 9 males and 47 females, aged 18–22 years. Inclusion criteria were: 1. aged 18–24 years, 2. enrolled at a British university in mainland China (foundation year and undergraduate studies), 3. no prior participation in a formal mindfulness-based intervention, 4. strong English proficiency (Gaokao English ≥ 115, IELTS ≥ 5.5, or equivalent), and 5. ability and willingness to attend all group sessions and complete home assignments. Exclusion criteria were: 1. incomplete baseline personality data, and 2. failure to submit at least five entries per week in the daily meditation journal.

### 2.2. Intervention Design

Participants were divided into three groups: intervention Group 1 (*n* = 21), intervention Group 2 (*n* = 18), and a waiting list control group (*n* = 17). Each eight-week intervention included eight two-hour, face-to-face group sessions, together with daily formal and informal meditation practice (averaging 30 min per day), which participants recorded in a meditation journal. Each session began with a reflection on the previous week’s formal and informal meditation practice, followed by an interactive seminar on the week’s topic, and concluded with a guided group meditation. Group 1 participated in an MBI incorporating concentration- and ethics-based meditation practices (MBI-CE), while Group 2 participated in an MBI incorporating concentration-, ethics-, and wisdom-based meditation practices (MBI-CEW). The two MBIs were conducted concurrently and divided into three stages: during Stage 1 (two-weeks), both MBIs engaged in concentration-based practices; during Stage 2 (three-weeks), both engaged in ethics-based practices; and during Stage 3 (three-weeks), MBI-CE continued with ethics-based practices, while MBI-CEW transitioned to wisdom-based practices.

More specifically, during Stage 1 participants engaged in (1) mindfulness of breath, and (2) mindfulness of body. Subsequently, during Stage 2 they focussed on (3) mindfulness of feelings, (4) observing the quality of the mind, and (5) loving-kindness meditation. During Stage 3, the two interventions diverged with the MBI-CE participants exploring practices such as (6a) compassion meditation, (7a) sympathetic-joy meditation, and (8a) equanimity meditation, while the MBI-CEW participants practiced (6b) meditation on impermanence, (7b) meditation on interdependence, (8b) meditation on emptiness and no-self. The meditations used in the two interventions were adapted from established MBIs and meditation programmes, drawing specifically on practices from Meditation Awareness Training (MAT; [41]), the Barre Center for Buddhist Studies’ Mindfulness of Breathing course, and Sharon Salzberg’s teachings on the Buddhist Four Immeasurables: lovingkindness, compassion, sympathetic joy, and equanimity.

Participants were asked to complete at least five entries per week in their daily meditation journals, documenting both formal and informal practice. These journals were submitted to the facilitator at the start of each session as a manipulation check to verify engagement with the assigned techniques. Participants who did not meet this requirement were deemed to have not completed the course and were excluded from the final dataset.

Both the MBI-CE and MBI-CEW were administered by the same facilitator (and primary author) who is trained in MBSR, Meditation Awareness Training, the Buddhist four foundations of mindfulness (*Satipaṭṭhāna*) practice and Buddhist mindfulness of breathing (*Ānāpānasati)* practice, as well as holding a *NeuroMindfulness* coaching certificate accredited by the International Coaching Federation.

### 2.3. Measures

#### 2.3.1. Ten-Item Personality Inventory

The Ten-Item Personality Inventory (TIPI; ref. [42]) was used to measure participants’ Big Five personality traits. Designed as a brief alternative to more extensive five-factor model instruments, the TIPI has been utilised in at least 29 academic studies [43]. It demonstrates acceptable test–retest reliability, patterns of predicted external correlates, convergence between self and observer rating, and convergence with widely used Big-Five measures in self, observer, and peer reports [42,43]. Each Big Five trait is assessed using two items (one positively worded and one negatively worded). For example, items for Extraversion include statements such as “I see myself as: extraverted, enthusiastic” and “I see myself as: reserved, quiet”. Responses are scored on a seven-point Likert scale (ranging from “*disagree strongly*” to “*agree strongly*”). The TIPI demonstrated acceptable two-item reliability for Extraversion (r = 0.71, ρSB = 0.83). The remaining traits also showed estimates that, while lower than those typically observed with longer Big Five inventories, are appropriate for a brief two-item-per-trait measure [42]: Agreeableness (r = 0.26, ρSB = 0.41), Conscientiousness (r = 0.60, ρSB = 0.75), Neuroticism (r = 0.55, ρSB = 0.71), and Openness (r = 0.37, ρSB = 0.54).

#### 2.3.2. Ryff’s Brief Scale of Psychological Well-Being

Ryff’s Brief Scale of Psychological Well-being (PWB; ref. [44]) was used to assess perceived psychological well-being of participants. This 18-item scale evaluates six dimensions of well-being including: autonomy, environmental mastery, personal growth, positive relations with others, purpose in life, and self-acceptance. Items include statements such as “The demands of everyday life often get me down” and “When I look at the story of my life, I am pleased with how things have turned out so far”. Responses are scored on a six-point Likert scale (ranging from “*strongly disagree*” to “*strongly agree*”). The PWB demonstrated modest internal consistency in the current study, with McDonald’s Omega reliability estimated at 0.66.

#### 2.3.3. Prosocialness Scale for Adults

The Prosocialness Scale for Adults (PSA; ref. [45]) was used to measure individual self-reported prosocial tendencies. This 16-item scale evaluates four types of prosocial actions: sharing, helping, caregiving, and empathising with others’ needs or requests. Items include statements such as “I am pleased to help my friends/colleagues in their activities” and “I am available for volunteer activities to help those who are in need”. Responses are scored on a five-point Likert scale (ranging from “*never true*” to “*always true*”) with higher scores indicating greater prosocialness. The PSA demonstrated excellent internal consistency in the current study, with McDonald’s Omega reliability estimated at 0.94.

#### 2.3.4. Intervention Delivery Checks

To minimise facilitator bias toward the MBI-CE or MBI-CEW, participants evaluated both the instructor and the intervention at the end of the course. The Student Evaluation of Teacher (SET) included a 5-item scale with statements such as “The instructor was well prepared for each class.” Similarly, the Student Evaluation of Intervention (SEI) consisted of a 5-item scale with items like “The course was organized in a manner that helped me understand the concepts and meditation practices.” Both evaluations used a five-point Likert scale, ranging from “*Strongly Agree*” to “*Strongly Disagree*.” Additionally, each evaluation included a final item to assess overall satisfaction: the SET asked about satisfaction with the instructor, and the SEI asked about satisfaction with the course, scored on a five-point Likert scale from “*Extremely Satisfied*” to “*Extremely Unsatisfied*”.

### 2.4. Procedure

Participants were recruited over a one-month period (September 2023) prior to the start of the interventions. Recruitment took place at a British university in mainland China and targeted foundation year and undergraduate students. Recruitment methods included online flyers, an introductory presentation at an extracurricular module fair, and promotion on the university’s career services website. Participants first completed demographic questions and were then randomly assigned to an intervention group. They subsequently completed the TIPI to assess personality traits, followed by the PSA to measure prosocial attitudes and the PWB to assess psychological well-being at baseline and again at the end of the intervention.

### 2.5. Statistical Analysis

Personality traits were treated as continuous variables based on participants’ scores on the Ten-Item Personality Inventory (TIPI). This approach preserves the full variability of the data and avoids the loss of information associated with dichotomizing continuous variables [46]. Relationships between personality traits and intervention outcomes were evaluated using a two-step framework.

First, intervention effects and potential moderation by personality traits were examined using a linear regression framework. The primary analysis modelled post-intervention scores as the outcome, with baseline scores included as covariates. Intervention Group (Control, MBI-CE, MBI-CEW), Personality Trait (continuous), and their interaction (Group × Personality Trait) were included as predictors. The interaction term tested whether the effect of the intervention on outcomes was moderated by personality traits. Effect sizes for the regression models are reported as standardised beta coefficients (β) and R^2^ values.

Second, for significant interactions between intervention group and personality traits, simple slopes analyses were performed to examine the direction and strength of the relationship between intervention group and outcomes at different levels of the personality trait (e.g., ±1 SD from the mean). Interaction plots were created to visually represent these effects for significant findings. The points for the interaction plots were calculated using the regression equation derived from the linear model, incorporating the coefficients for main effects and interaction terms. Predicted well-being scores were computed for each intervention group at three levels of Openness: Low (−1 SD), Mean (0 SD), and High (+1 SD).

To ensure the robustness of findings, a sensitivity analysis using an ANCOVA framework was conducted. In this approach, intervention Group (Control, MBI-CE, MBI-CEW) and Personality Trait Level (Low, High, based on a median split) were treated as fixed factors. The interaction term (Group × Trait Level) tested whether the intervention effects differed across personality levels. While ANCOVA assumptions were considered, their strict adherence was less critical in this context, as the analysis served as a secondary sensitivity check rather than the primary analytic approach. This complementary analysis provided additional evidence for the moderating role of personality traits, with effect sizes reported as partial η^2^ for overall tests and Hedges’ g for pairwise comparisons, further validating the observed patterns. To clarify subgroup patterns, a small set of planned simple-effects contrasts was conducted within each trait level (e.g., within High: MBI-CEW vs. MBI-CE and MBI-CEW vs. Control), using change scores (Post–Pre) for interpretability. Each contrast is reported with the estimated mean difference, 95% confidence interval, *p*-value, and standardised effect size (Hedges’ g).

However, the continuous-variable approach was prioritised for its methodological rigour and ability to capture individual differences more precisely.

## 3. Results

### 3.1. Descriptive Statistics

Descriptive statistics are presented prior to hypothesis testing. Table 1 reports the means and standard deviations for prosocial attitudes (PSA) and psychological well-being (PWB) at baseline (pre-) and post-intervention, categorised by group. Additionally, descriptive statistics for trait and outcome variables are summarised in Table 2.

### 3.2. Moderation Effects

The following results are presented in two sections: first, the moderation effects of personality traits on prosocial outcomes, followed by their effects on well-being. For each personality trait, linear regression models were used to test Group × Personality Trait interactions, with baseline scores included as covariates. Linear regression results for prosocial and well-being outcomes are presented in Table 3 and Table 4, respectively. When a Group × Personality Trait interaction was found to be significant, simple slopes analyses were conducted to examine the direction and strength of the moderation effect at different levels of the personality trait (e.g., ±1 SD from the mean).

To complement the regression findings and validate the moderating role of personality traits, a sensitivity analysis using an ANCOVA framework was also performed (see Appendix A for descriptive statistics). Additionally, planned contrasts were conducted to further explore these interactions, providing a more detailed understanding of the moderation effects (see Appendix A for detailed results).

### 3.3. Personality Trait Moderation of Prosocialness

#### 3.3.1. Neuroticism

The linear regression model for Neuroticism explained 57.8% of the variance in post-intervention prosocial scores (R^2^ = 0.578, F(6,49) = 11.17, *p* < 0.001). Baseline prosocial scores were a significant predictor of post-intervention scores (β = 0.678, *p* < 0.001), indicating that higher pre-intervention prosocial scores were associated with higher post-intervention scores.

The main effects of intervention group (MBI-CE: β = −6.71, *p* = 0.317; MBI-CEW: β = −13.44, *p* = 0.079) and Neuroticism (β = −0.68, *p* = 0.240) were not significant. However, the interaction between MBI-CEW and Neuroticism was significant (β = 2.09, *p* = 0.021), suggesting that the effect of MBI-CEW on prosocial outcomes varied depending on Neuroticism levels. The interaction between MBI-CE and Neuroticism was not significant (β = 0.70, *p* = 0.372).

To further explore the significant interaction between intervention group and Agreeableness, simple slopes analyses were conducted. At High Neuroticism (+1 SD), the interaction between MBI-CEW and Neuroticism was significant (β = 5.55, SE = 2.33, t = 2.38, *p* = 0.021), indicating that MBI-CEW produced significantly greater improvements in prosocial scores compared to both MBI-CE and Control. In contrast, at Mean Neuroticism (0 SD), the interaction effect was not significant (β = 3.99, SE = 2.20, t = 1.82, *p* = 0.075), and at Low Neuroticism (−1 SD), the interaction effect was also not significant (β = −1.60, SE = 3.09, t = −0.52, *p* = 0.607). These results (represented in Figure 1) suggest that the intervention effects of MBI-CEW were particularly pronounced for individuals with higher levels of Neuroticism, while no such effects were observed for those with average or lower levels of the trait.

Consistent with the regression model, the ANCOVA sensitivity analysis revealed a significant Group × Neuroticism interaction, F(2,49) = 3.37, *p* = 0.043, partial η^2^ = 0.121. The planned contrasts on change scores also showed that, among participants high in Neuroticism (see Figure 2), MBI-CEW produced significantly greater prosocial gains than both MBI-CE (Δ = 13.67, 95% CI [7.02, 20.31], *p* < 0.001, Hedges’ g = 1.69) and Control (Δ = 11.13, 95% CI [7.53, 14.72], *p* < 0.001, Hedges’ g = 2.86). Among low-Neuroticism participants (see Figure 3), group differences were small and non-significant (all *p* ≥ 0.52, g ≤ 0.30).

#### 3.3.2. Agreeableness

The linear regression model for Agreeableness explained 53.6% of the variance in post-intervention prosocial scores (R^2^ = 0.536, F(6,49) = 9.44, *p* < 0.001). Baseline prosocial scores were a significant predictor of post-intervention scores (β = 0.656, *p* < 0.001), indicating that higher pre-intervention prosocial scores were associated with higher post-intervention scores.

Neither the main effects of intervention group (MBI-CE: β = −1.33, *p* = 0.545; MBI-CEW: β = 4.03, *p* = 0.086) nor Agreeableness (β = 0.814, *p* = 0.614) were statistically significant. Additionally, the interaction terms (Group × Agreeableness) were not significant (MBI-CE × Agreeableness: β = −0.879, *p* = 0.675; MBI-CEW × Agreeableness: β = 0.866, *p* = 0.721), suggesting that Agreeableness did not moderate the effect of the intervention on prosocial outcomes.

Consistent with the regression model, the ANCOVA sensitivity analysis found no significant Group × Openness interaction, F(2,49) = 0.23, *p* = 0.799, partial η^2^ = 0.009. Planned contrasts revealed that among high-Agreeableness participants, MBI-CEW did not differ significantly from either MBI-CE (Δ = 7.20, 95% CI [−3.90, 18.29], *p* = 0.184, Hedges’ g = 0.68) or Control (Δ = 5.51, 95% CI [−3.14, 14.17], *p* = 0.183, Hedges’ g = 0.70). Among low-Agreeableness participants, MBI-CEW did not differ significantly from Control (Δ = 5.23, 95% CI [−0.98, 11.43], *p* = 0.092, Hedges’ g = 0.84) and showed only a modest, non-significant advantage over MBI-CE (Δ = 6.24, 95% CI [0.06, 12.42], *p* = 0.048, Hedges’ g = 0.77). These findings suggest that Openness did not moderate the effects of the interventions (MBI-CE and MBI-CEW) on prosocial outcomes.

#### 3.3.3. Conscientiousness

Similarly, the linear regression model for Conscientiousness explained 59.5% of the variance in post-intervention prosocial scores (R^2^ = 0.595, F(6,49) = 12.00, *p* < 0.001). Baseline prosocial scores were a significant predictor of post-intervention scores (β = 0.661, *p* < 0.001), indicating that higher pre-intervention prosocial scores were associated with higher post-intervention scores.

Neither the main effects of intervention group (β = −10.46, *p* = 0.101; MBI-CEW: β = −2.66, *p* = 0.747) nor Conscientiousness (β = −0.084, *p* = 0.896) were statistically significant. Additionally, the interaction terms (Group × Conscientiousness) were not significant (MBI-CE × Conscientiousness: β = 1.15, *p* = 0.132; MBI-CEW × Conscientiousness: β = 0.79, *p* = 0.423), suggesting that Conscientiousness did not moderate the effect of the intervention on prosocial outcomes.

The ANCOVA sensitivity analysis yielded results consistent with the regression model, showing no significant Group × Openness interaction, F(2,49) = 1.43, *p* = 0.250, partial η^2^ = 0.055. However, planned contrasts revealed that among low-Conscientiousness participants, MBI-CEW showed significantly greater gains in prosocial scores compared to MBI-CE (Δ = 9.24, 95% CI [3.18, 15.31], *p* = 0.005, g = 1.21) and Control (Δ = 4.88, 95% CI [0.38, 9.37], *p* = 0.036, g = 1.11). Among high-Conscientiousness participants, no significant differences were observed (MBI-CEW vs. MBI-CE: Δ = 2.15, 95% CI [−6.79, 11.09], *p* = 0.617, g = 0.23; MBI-CEW vs. Control: Δ = 5.62, 95% CI [−2.34, 13.58], *p* = 0.154, g = 0.65). These findings suggest potential advantages for MBI-CEW among low-Conscientiousness participants, though interpretation is limited by the non-significant omnibus interaction and unbalanced cell sizes.

#### 3.3.4. Openness

Furthermore, the linear regression model for Openness explained 54.9% of the variance in post-intervention prosocial scores (R^2^ = 0.549, F(6,49) = 9.92, *p* < 0.001). Baseline prosocial scores were a significant predictor of post-intervention scores (β = 0.634, *p* < 0.001), indicating that higher pre-intervention prosocial scores were associated with higher post-intervention scores.

Neither the main effects of intervention group (MBI-CE: β = −10.41, *p* = 0.455; MBI-CEW: β = −5.32, *p* = 0.674) nor Openness (β = −0.28, *p* = 0.783) were statistically significant. Additionally, the interaction terms (Group × Openness) were not significant (MBI-CE × Openness: β = 0.89, *p* = 0.481; MBI-CEW × Openness: β = 0.99, *p* = 0.392), suggesting that Openness did not moderate the effect of the intervention on prosocial outcomes.

Consistent with the regression model, the ANCOVA sensitivity analysis found no significant Group × Openness interaction, F(2,50) = 0.39, *p* = 0.677, partial η^2^ = 0.015. However, planned contrasts revealed that among high-Openness participants, MBI-CEW produced significantly greater gains in prosocial scores compared to MBI-CE (Δ = 11.65, 95% CI [5.04, 18.26], *p* = 0.002, g = 1.32) and Control (Δ = 9.95, 95% CI [4.53, 15.37], *p* = 0.002, g = 1.81). Among low-Openness participants, no significant differences emerged (all *p* ≥ 0.57), and comparisons with Control could not be made due to a zero cell in that subgroup. These findings suggest potential advantages for MBI-CEW among high-Openness participants, though interpretation is limited by unbalanced cell sizes.

#### 3.3.5. Extraversion

Finally, the linear regression model for Extraversion explained 54.9% of the variance in post-intervention prosocial scores (R^2^ = 0.549, F(6,49) = 9.95, *p* < 0.001). Baseline prosocial scores were a significant predictor of post-intervention scores (β = 0.653, *p* < 0.001), indicating that higher pre-intervention prosocial scores were associated with higher post-intervention scores.

Neither the main effects of intervention group (MBI-CE: β = −4.59, *p* = 0.450; MBI-CEW: β = 3.82, *p* = 0.524) nor extraversion (β = −0.48, *p* = 0.312) were statistically significant. Additionally, the interaction terms (Group × Extraversion) were not significant (MBI-CE × Extraversion: β = 0.34, *p* = 0.623; MBI-CEW × Extraversion: β = −0.09, *p* = 0.897), suggesting that extraversion did not moderate the effect of the intervention on prosocial outcomes.

The ANCOVA sensitivity analysis yielded results consistent with the regression model, showing no significant Group × Openness interaction, F(2,49) = 0.21, *p* = 0.811, partial η^2^ = 0.009. However, exploratory planned contrasts revealed that among low-extraversion participants, MBI-CEW showed significantly greater gains in prosocial scores compared to MBI-CE (Δ = 8.58, 95% CI [1.91, 15.26], *p* = 0.014, g = 0.99) and a strong trend relative to Control (Δ = 6.17, 95% CI [−0.10, 12.45], *p* = 0.053, g = 0.90). No significant differences were observed among high-extraversion participants, with MBI-CEW not differing significantly from either MBI-CE (Δ = 3.57, 95% CI [−6.71, 13.85], *p* = 0.464, g = 0.38) or Control (Δ = 3.55, 95% CI [−4.82, 11.92], *p* = 0.351, g = 0.51).

These findings suggest that while extraversion did not significantly moderate intervention effects in the primary regression analysis, exploratory ANCOVA results indicate that low-extraversion participants may benefit more from the MBI-CEW intervention compared to other groups.

### 3.4. Personality Trait Moderation of Well-Being

#### 3.4.1. Neuroticism

The linear regression model for Neuroticism explained 85.2% of the variance in post-intervention well-being scores (R^2^ = 0.852, F(6,49) = 46.98, *p* < 0.001). Baseline well-being scores were a significant predictor of post-intervention scores (β = 0.387, *p* = 0.001), indicating that individuals with higher pre-intervention well-being scores tended to report higher post-intervention scores.

The main effects of intervention group were not significant for either MBI-CE (β = 7.521, *p* = 0.343) or MBI-CEW (β = 6.751, *p* = 0.446), suggesting that the interventions did not independently predict post-intervention well-being when controlling for Neuroticism and baseline well-being. Neuroticism, however, had a small but significant negative main effect (β = −1.648, *p* = 0.010), indicating that individuals with higher Neuroticism scores tended to report slightly lower post-intervention well-being. The interaction between MBI-CE and Neuroticism (β = 1.520, *p* = 0.068) and MBI-CEW and Neuroticism (β = 1.671, *p* = 0.076) did not meet conventional thresholds for statistical significance.

Consistent with the regression model, the ANCOVA sensitivity analysis found no significant Group × Neuroticism interaction, F(2,49) = 1.94, *p* = 0.155, partial η^2^ = 0.073. Planned contrasts on change scores showed no significant differences among high-Neuroticism participants: MBI-CEW did not differ significantly from MBI-CE (Δ = 1.03, 95% CI [−6.00, 8.06], *p* = 0.759, g = 0.13) or Control (Δ = 5.49, 95% CI [−1.33, 12.30], *p* = 0.105, g = 0.68). Similarly, among low-Neuroticism participants, differences were negligible (all *p* ≥ 0.339, g ≤ 0.43). These findings suggest that Neuroticism did not moderate the effects of the interventions (MBI-CE and MBI-CEW) on well-being outcomes, and the observed effects were consistent across levels of Neuroticism.

#### 3.4.2. Agreeableness

The linear regression model for Agreeableness explained 85.4% of the variance in post-intervention well-being scores (R^2^ = 0.854, F(6,49) = 47.93, *p* < 0.001). Baseline well-being scores were a significant predictor of post-intervention scores (β = 0.343, *p* = 0.002), indicating that individuals with higher pre-intervention well-being scores tended to report higher post-intervention scores.

The main effects of intervention group were not significant for either MBI-CE (β = 6.401, *p* = 0.403) or MBI-CEW (β = 0.879, *p* = 0.919). Agreeableness, however, had a small but significant negative main effect (β = −1.775, *p* = 0.012), indicating that individuals with higher Agreeableness scores tended to report slightly lower post-intervention well-being. The interaction between MBI-CE and Agreeableness was significant (β = 1.873, *p* = 0.044), as well as the interaction between MBI-CEW and Agreeableness (β = 2.701, *p* = 0.012), indicating that the effect of both the MBI-CE and MBI-CEW on well-being were moderated by Agreeableness.

To further explore the significant interaction between intervention group and Agreeableness, simple slopes analyses were conducted. At High Agreeableness (+1 SD), the interaction between MBI-CE and Agreeableness was significant (β = 23.75, SE = 2.16, t = 11.02, *p* < 0.001), indicating that the effect of MBI-CE on well-being was stronger for individuals with high Agreeableness. Similarly, the interaction between MBI-CEW and Agreeableness was significant (β = 18.21, SE = 2.16, t = 8.43, *p* < 0.001), suggesting that individuals with high Agreeableness benefited more from the MBI-CEW intervention. At Mean Agreeableness (0 SD), the interaction effects were significant for both MBI-CE (β = 20.82, SE = 2.16, t = 9.64, *p* < 0.001) and MBI-CEW (β = 21.67, SE = 2.16, t = 10.03, *p* < 0.001). However, at Low Agreeableness (−1 SD), the interaction effects were not significant for MBI-CE (β = 1.67, SE = 12.93, t = 0.13, *p* = 0.905) or MBI-CEW (β = −6.87, SE = 12.93, t = −0.53, *p* = 0.600). These results (represented in Figure 4) suggest that the intervention effects of MBI-CE and MBI-CEW were more pronounced for individuals with higher Agreeableness, while those with low Agreeableness did not experience significant variation in well-being outcomes based on the interventions.

Consistent with the findings from the linear regression model, the ANCOVA sensitivity analysis for Agreeableness revealed a significant Group × Agreeableness interaction (F(2,49) = 3.60, *p* = 0.035, partial η^2^ = 0.128). This indicates that the relationship between intervention group and post-intervention well-being varied by Agreeableness levels. However, planned contrasts did not reveal significant pairwise differences within either high- or low-Agreeableness subgroups (all *p* ≥ 0.441, g ≤ 0.41).

#### 3.4.3. Conscientiousness

The linear regression model examining the role of Conscientiousness explained 84.2% of the variance in post-intervention well-being scores ($R^{2}=0.842$, F(6,49)=43.43, p<0.001). Baseline well-being scores were a significant predictor of post-intervention scores (β=0.413, p<0.001), indicating that individuals with higher pre-intervention well-being scores tended to report higher post-intervention scores.

The main effects of intervention group were significant, with both MBI-CE (β=28.24, p=0.001) and MBI-CEW (β=34.31, p=0.001) demonstrating substantial improvements in well-being compared to the Control group. Conscientiousness, however, did not have a significant main effect (β=0.891, p=0.231), suggesting that Conscientiousness alone was not a strong predictor of post-intervention well-being. The interaction between MBI-CE and Conscientiousness was not significant (β=−1.260, p=0.148), indicating that the effect of MBI-CE on well-being did not vary meaningfully across levels of Conscientiousness. Additionally, although trending toward significance the interaction between MBI-CEW and Conscientiousness did not meet conventional thresholds for statistical significance (β=−1.916, p=0.093).

However, the ANCOVA sensitivity analysis provided additional insights, revealing a significant Group × Conscientiousness interaction (F(2,49) = 3.43, *p* = 0.040, partial η^2^ = 0.123). Planned contrasts showed that among low-Conscientiousness participants MBI-CEW significantly improved well-being compared to the Control group (Δ = 7.63, 95% CI [1.98, 13.27], *p* = 0.0129, g = 1.42). No other subgroup comparisons reached significance. These findings suggest that while the overall interaction effects were not robust in the regression model, the ANCOVA results highlight a potential benefit of MBI-CEW for individuals with low-Conscientiousness.

#### 3.4.4. Openness

The linear regression model for Openness explained 85.4% of the variance in post-intervention well-being scores ($R^{2}=0.854$, F(6,49)=47.74, p<0.001). Baseline well-being scores were a significant predictor of post-intervention scores (β=0.335, p=0.003), indicating that individuals with higher pre-intervention well-being scores tended to report higher post-intervention scores.

The main effects of intervention group were not significant for either MBI-CE (β=4.999, p=0.720) or MBI-CEW (β=−6.871, p=0.600), suggesting that the interventions did not independently predict post-intervention well-being when controlling for Openness and baseline well-being. Openness, however, had a small but significant negative main effect (β=−2.373, p=0.026), indicating that individuals with higher Openness scores tended to report slightly lower post-intervention well-being.

The interaction between MBI-CE and Openness was not significant (β=1.192, p=0.342), suggesting that the effect of MBI-CE on well-being did not vary meaningfully across levels of Openness. In contrast, the interaction between MBI-CEW and Openness was significant (β=2.478, p=0.045), indicating that the effect of MBI-CEW on well-being was moderated by Openness.

To further explore the significant interaction between intervention group and Openness, simple slopes analyses were conducted. At High Openness (+1 SD), the interaction between MBI-CEW and Openness was significant (β=2.478, SE = 1.207, t=2.054, p=0.045), suggesting that MBI-CEW produced greater improvements in well-being compared to the Control group. However, the interaction between MBI-CE and Openness was not significant (β=1.192, SE = 1.241, t=0.960, p=0.342). At Mean Openness (0 SD), the main effects of MBI-CE (β=4.999, p=0.720) and MBI-CEW (β=−6.871, p=0.600) were not significant, indicating that neither intervention significantly improved well-being compared to the Control group at average levels of Openness. At Low Openness (−1 SD), the interaction effects were not significant for either MBI-CE (β=1.192, p>0.05) or MBI-CEW (β=2.478, p>0.05). These results (represented in Figure 5) suggest that the intervention effects of MBI-CEW were particularly pronounced for individuals with higher levels of Openness, while no such effects were observed for those with average or lower levels of the trait.

In contrast, the ANCOVA sensitivity analysis did not support these findings. The Group × Openness interaction was not significant (F(2,50)=1.40, p=0.257, partial $\eta^{2}=0.053$), indicating no meaningful moderation effect of Openness on intervention outcomes. Within high-Openness participants, MBI-CEW did not differ significantly from either MBI-CE (Δ=2.49, 95% CI [−4.96, 9.94], p=0.464, g=0.40) or the Control group (Δ=1.02, 95% CI [−7.91, 9.94], p=0.813, g=0.09). Thus, the interaction effect for MBI-CEW at high Openness observed in the linear regression should be interpreted with caution.

#### 3.4.5. Extraversion

The linear regression model examining the role of Extraversion explained 84.5% of the variance in post-intervention well-being scores ($R^{2}=0.845$, F(6,49)=44.37, p<0.001). Baseline well-being scores were a significant predictor of post-intervention scores (β=0.425, p<0.001), indicating that individuals with higher pre-intervention well-being scores tended to report higher post-intervention scores.

The main effects of intervention group were significant, with both MBI-CE (β=17.02, p<0.001) and MBI-CEW (β=17.36, p<0.001) demonstrating substantial improvements in well-being compared to the Control group. Extraversion, however, had a small but significant negative main effect (β=−3.29, p=0.049), suggesting that individuals with higher Extraversion scores tended to report slightly lower post-intervention well-being. The interaction between MBI-CE and Extraversion, although trending toward significance, did not meet conventional thresholds for statistical significance (β=4.56, p=0.063). Likewise, the interaction between MBI-CEW and Extraversion was not significant (β=3.61, p=0.130).

To further explore the interaction between intervention group and Extraversion, simple slopes analyses were conducted. At High Extraversion (+1 SD), the interaction between MBI-CEW and Extraversion was not significant (β = 1.097, SE = 0.713, t = 1.540, *p* = 0.130), and the interaction between MBI-CE and Extraversion trended toward significance (β = 1.389, SE = 0.730, t = 1.903, *p* = 0.063). At Mean Extraversion (0 SD), the main effects of MBI-CE (β = 5.960, *p* = 0.445) and MBI-CEW (β = 8.623, *p* = 0.267) were not significant, indicating that neither intervention significantly improved well-being compared to the Control group at average levels of Extraversion. At Low Extraversion (−1 SD), the interaction effects were not significant for either MBI-CE (β = 1.097, *p* > 0.05) or MBI-CEW (β = 1.389, *p* > 0.05). These results suggest that the intervention effects of MBI-CE and MBI-CEW were not significantly influenced by levels of Extraversion.

The ANCOVA sensitivity analysis also revealed that the Group × Extraversion interaction was not significant (F(2,49)=0.55, p=0.579, partial $\eta^{2}=0.022$), and change-score contrasts showed no meaningful differences in either high- or low-Extraversion subgroups (p≥0.422, g≤0.42). These findings suggest that Extraversion did not moderate the effects of the interventions (MBI-CE and MBI-CEW) on well-being outcomes.

### 3.5. Intervention Delivery Checks

Results of a univariate two-way ANOVA revealed no statistically significant differences between MBI-CE and MBI-CEW participants’ evaluation of the instructor’s preparedness for the course [F(1,37) = 0.530, *p* = 0.471, ηp^2^ = 0.014] or the course organisation [F(1,37) = 2.748, *p* = 0.106, ηp^2^ =0.069]. Likewise, there were no statistically significant differences between MBI-CE and MBI-CEW participants’ overall satisfaction with the instructor [F(1,37) = 0.450, *p* = 0.506, ηp^2^ =0.012] or course [F(1,37) = 0.749, *p* = 0.392, ηp^2^ =0.020]. These findings suggest that participants in both the MBI-CE and MBI-CEW groups perceived the instructor and course delivery similarly, with no significant differences in their evaluations.

## 4. Discussion

This study investigated how the Big Five personality traits moderate the effects of two types of MBIs on psychological well-being and prosocial behaviour. Participants were assigned either to an intervention including concentration- and ethics-based meditation practices (MBI-CE), to an intervention incorporating concentration-, ethics-, and wisdom-based meditation practices (MBI-CEW), or to a control group. The primary aim was to assess whether the inclusion of ethics-based practices, or the combination of ethics- and wisdom-based practices, enhanced outcomes for individuals with different personality traits, while also exploring whether these practices mitigated any unintended negative effects.

### 4.1. Hypothesis 1: Openness to Experience

It was hypothesised that high openness would positively moderate the effects of the MBI-CEW intervention, but not the MBI-CE, across both well-being and prosocial behaviour outcomes. For well-being, a significant interaction was observed between openness and the MBI-CEW intervention, which was not present in the MBI-CE or control groups. Notably, this interaction was specific to high levels of openness and was not observed at lower levels of the trait. For prosocial behaviour, no significant interaction was found in the primary linear regression analysis. However, planned contrasts from the ANCOVA sensitivity analysis suggested that, among high-openness participants, the MBI-CEW intervention may have produced greater gains in prosocial scores compared to the MBI-CE intervention, though this finding warrants cautious interpretation.

These findings tentatively support the hypothesis that high openness is particularly beneficial for individuals engaging in MBIs that include both ethics- and wisdom-based practices for well-being and prosocial outcomes, while it does not appear be a necessary factor for MBIs only incorporating ethics-based practices.

### 4.2. Hypothesis 2: Conscientiousness

It was hypothesised that high conscientiousness would positively moderate the effects of both the MBI-CE and MBI-CEW, across well-being and prosocial behaviour outcomes. However, for well-being, no significant interaction between conscientiousness and either intervention was observed in the primary linear regression analysis. Similarly, no significant interaction was found for prosocial behaviour.

These findings do not support the hypothesis that high conscientiousness moderates the effects of the interventions. This hypothesis assumed that conscientious individuals, characterised by self-discipline, organisation, and a strong sense of responsibility [29], would engage more effectively with the meditation practices included in the interventions. However, the lack of significant findings may be explained by the inclusion of manipulation checks, such as required daily meditation journaling, which were designed to verify participants’ engagement with the assigned techniques.

Conscientiousness is typically considered an internal motivator, driving individuals to engage in interventions through their intrinsic sense of responsibility and discipline [47]. In contrast, manipulation checks such as a daily journal can act as external motivators by creating accountability and encouraging engagement regardless of personality traits [48]. By increasing external motivation, these mechanisms may reduce the reliance on conscientiousness as an internal driver of adherence, potentially explaining why conscientiousness did not significantly moderate the intervention effects in this study.

### 4.3. Hypothesis 3: Extraversion

It was hypothesised that high Extraversion would negatively moderate the effects of the MBI-CEW intervention, but not the MBI-CE intervention, on well-being and prosocial behaviour outcomes. For well-being, no significant interaction between intervention group and Extraversion was observed at any level of the trait, indicating that the effects of the interventions on well-being were not influenced by Extraversion. Similarly, for prosocial outcomes the linear regression analysis did not identify Extraversion as a significant moderator of intervention effects. However, exploratory planned contrasts revealed an intriguing pattern: participants with low Extraversion experienced significantly greater gains in prosocial outcomes from the MBI-CEW intervention compared to the MBI-CE intervention. While noteworthy, this finding should be interpreted with caution, as it was exploratory and not part of the primary analysis.

Overall, the findings do not support the hypothesis that high Extraversion moderates the effects of the MBI-CEW intervention. This hypothesis was based on the assumption that wisdom-based practices were inherently slower-paced and more interoceptive than ethics-based practices, and thus less suited for highly extraverted individuals [5]. However, this assumption may not hold true, as a core component of most contemplative practices, regardless of their specific focus, involves a degree of interoceptive awareness [49]. Additionally, as both interventions included face-to-face sessions, the interoceptive nature of meditation practices, which might otherwise pose challenges for individuals with high Extraversion, may have been offset by the opportunities for interpersonal interactions provided during these sessions [1].

### 4.4. Hypothesis 4: Agreeableness

It was hypothesised that high Agreeableness would positively moderate the effects of both the MBI-CE and MBI-CEW interventions across well-being and prosocial outcomes. For well-being, linear regression analysis revealed a significant interaction between Agreeableness and both interventions. Simple slopes analysis confirmed that the benefits of the interventions were more pronounced for individuals with high Agreeableness. However, no such interaction was found for prosocial outcomes, suggesting that Agreeableness did not moderate the effects of the interventions in this domain.

Although these findings support the hypothesis that high Agreeableness moderates the effects of both the MBI-CE and MBI-CEW for well-being, they do not support their influence over prosocial outcomes. This discrepancy may stem from the inherent complexity of assessing prosocial behaviour as an outcome, given its multilevel nature [50]. Prosocial behaviour can be understood across three distinct levels: the micro level, which examines the origins of prosocial tendencies within individuals; the meso level, which focuses on helper-recipient dyads and interpersonal interactions; and the macro level, which considers prosocial actions within the broader context of groups and communities [50]. The Prosocialness Scale for Adults [45] used within this study only measured at the micro level though individual self-reported prosocial tendencies which may not fully capture the broader, situational, or group-based aspects of prosocial behaviour. As a result, the scale’s focus on individual tendencies might have limited its ability to detect changes influenced by the interventions at the meso or macro levels. Future research could benefit from incorporating measures that assess prosocial behaviour across all three levels to provide a more comprehensive understanding of how interventions impact this multilevel construct.

### 4.5. Hypothesis 5: Neuroticism

It was hypothesised that high Neuroticism would positively moderate the effects of both the MBI-CE and MBI-CEW interventions on well-being, while only moderating the effects of MBI-CEW on prosocial outcomes. For well-being, the findings revealed that Neuroticism did not significantly moderate the effects of either MBI-CE or MBI-CEW. However, for prosocial outcomes the results suggested that the effects of the MBI-CEW intervention were particularly pronounced for individuals with higher levels of Neuroticism, while no such effects were observed for individuals with average or lower levels of the trait.

Although, these findings support the hypothesis that individuals high Neuroticism will benefit from engaging in MBIs that include both ethics- and wisdom-based practices for prosocial outcomes, they do not support high Neuroticism as a moderator for well-being during either MBI. To understand this discrepancy, it is important to consider the distinct mechanisms through which MBIs may influence well-being and prosocial outcomes. The hypothesis regarding the interaction between Neuroticism and well-being was grounded in previous research, which suggests that highly neurotic individuals often experience greater improvements in subjective well-being and related factors, such as anxiety and mental distress, due to starting at a lower baseline [17,38]. However, in the present study, baseline well-being scores did not significantly differ between low- and high-neurotic individuals across the MBI-CE, MBI-CEW, and control groups (see Appendix A). This finding challenges the assumption that high-neurotic individuals were starting from a lower baseline, which may have implications for interpreting the results.

This discrepancy is noteworthy, as prior studies have consistently demonstrated strong associations between neuroticism and negative emotional states [50,51,52]. One possible explanation for this inconsistency could be the use of Ryff’s Scale of Psychological Well-being in the current study. Ryff’s scale evaluates well-being across six core dimensions: autonomy, environmental mastery, personal growth, purpose in life, positive relationships with others, and self-acceptance [44]. While this comprehensive approach provides a holistic view of well-being, it may not be sensitive enough to capture changes in the negative affect commonly experienced by high-neurotic individuals. As a result, the specific improvements in emotional distress or anxiety, which are often more pronounced in this population, may have been overlooked.

The supported hypothesis regarding the interaction between high Neuroticism and prosocial outcomes for the MBI-CEW is a significant finding that warrants further discussion. High neuroticism has been strongly linked to vulnerable narcissism, characterised by hypersensitivity and defensive self-focus [39]. Previous research suggests that during MBIs, individuals with high neuroticism may experience less pronounced benefits in prosocial attitudes, such as empathy, due to their heightened emotional reactivity and self-focus, which can interfere with the relational and outward-oriented aspects of mindfulness practices [18].

This may explain why different forms of meditation vary in their impact on prosocial outcomes, as vulnerable narcissism tends to fluctuate across interpersonal situations [53]. For example, ethics-based practices like loving-kindness meditation are explicitly relational, fostering warmth, friendliness, and care by directing attention toward others [54]. Evidence from contemporary reviews shows that these practices reliably enhance prosocial attitudes, such as empathy and connectedness, at a group level [55,56,57]. However, their focus on interpersonal relationships may unintentionally heighten interpersonal sensitivity in highly neurotic individuals prone to vulnerable narcissism, potentially amplifying defensiveness, coldness, and withdrawal [53].

In contrast, wisdom-based practices, such as contemplations on interdependence and emptiness, shift attention away from interpersonal evaluation toward a direct experiential recognition of oneness and transience. Research on emptiness and related wisdom meditations shows that these practices enhance non-attachment, compassion, and positive affect, while reducing rigid self-focus [58]. A systematic review of Buddhist-derived wisdom practices further supports their association with measurable increases in prosocial outcomes, likely due to their cultivation of a stable outward orientation [59]. Theoretically, this pattern can be explained as an erosion of ego-centric bias that supports the development of altruistic moral reasoning [60], and by self-transcendence (i.e., insight into interdependence) being the core mechanism linking meditation with prosocial behaviour [61]. By shifting the self-perception from a bounded, threatened ego to a sense of interconnectedness, wisdom practices reduce defensive self-protectiveness and enhance prosocial motivations [25,62]. This may explain why participants high in neuroticism, prone to vulnerable and self-protective states, showed the significant prosocial gains following the MBI-CEW but not the MBI-CE.

### 4.6. Ethics and/or Wisdom-Based Meditations

From the current study’s findings and the theories discussed above it appears that for well-being outcomes, conscientiousness, extraversion, and neuroticism did not moderate the effects of either the MBI-CE or MBI-CEW. However, high agreeableness moderated well-being outcomes for both interventions, while high openness moderated well-being only for the MBI-CEW. Furthermore, for prosocial outcomes, conscientiousness, extraversion, agreeableness and openness did not moderate the effects of either intervention. In contrast, high neuroticism moderated prosocial outcomes exclusively for the MBI-CEW, but not for the MBI-CE.

From these findings it can be interpreted that while high openness does not appear to be an important trait for engaging in concentration- and ethics-based meditation practices, it does appear to be significant when MBIs incorporate the addition of wisdom-based meditation practices. This may be due to the unfamiliarity of Buddhist wisdom-based practices and concepts, such as interdependence, non-self and emptiness, potentially requiring higher levels of openness to fully engage with them. To overcome this potential challenge for individuals with low openness, it is recommended that Buddhist wisdom concepts are introduced slowly and in relatable ways, building an analytical understanding before engaging in experiential contemplative practices.

Furthermore, findings suggest that wisdom-based meditation practices may be especially well suited to individuals high in neuroticism, for whom interpersonal sensitivity and vulnerable narcissistic states can undermine prosocial tendencies. This points toward the value of tailoring psychological interventions to personality profiles [11], aligning with the emerging need to better understand how personality moderates the effects of mindfulness training [17].

However, from the present findings, it is unclear if wisdom-based meditation practices alone are enough to benefit individuals high in neuroticism or if it was the combination on ethics and wisdom practices that led to these positive outcomes. One the one hand, critics have cautioned that the erosion of egoic boundaries cultivated in wisdom practices could, in some cases, risk being experienced as detachment if not grounded in compassion and ethical conduct [63]. For example, Buddhist teachings on no-self or non-attachment can be misinterpreted as pathological emptiness or disengagement when practitioners lack sufficient ego capacities or relational grounding. Thus, wisdom-based practices may appear to risk detachment, and potential reduction in prosocial attitudes, unless balanced by compassion and moral context found in ethics-based practices.

On the other hand, there are strong reasons to suggest that wisdom practices, even when delivered without explicit ethics-based components, can on their own cultivate prosocial attitudes. First, theoretical accounts of self-transcendence [64] and non-dual awareness [65] argue that dissolving rigid self-other boundaries naturally gives rise to feelings of unity and concern for others, rather than withdrawal. Supporting this, recent studies have found that participants primed with interdependence prior to a concentration-based meditation displayed significantly higher prosocial behaviour than those primed with independence, consistent with the idea that wisdom-oriented framings can steer attentional training toward connectedness and prosociality [66]. Similarly, Buddhist traditions describe wisdom as inseparable from compassion, with insight into interdependence naturally leading to care for others [67]. Overall, these perspectives suggest that wisdom-based practices, properly framed, may be sufficient to produce prosocial gains, even without the explicit addition of ethics-based practices, because they reconfigure the very foundations of self-other perception.

### 4.7. Limitations and Future Directions

This study is subject to several limitations, including the relatively small sample size that limits statistical power and restricts the generalizability of results. Similarly, the use of a non-clinical sample of undergraduate students may constrain applicability to clinical populations or to broader community samples outside of higher education. Furthermore, the absence of a long-term follow-up prevents conclusions about the durability of intervention effects over time. Future research could address these issues by recruiting larger and more diverse samples (i.e., 128 participants, 64 per group, to detect a medium effect size [Cohen’s d = 0.5] with 80% power and a significance level of 0.05) and by incorporating follow-up assessments three to six months post-intervention.

Additionally, the study relied exclusively on self-report measures. While this is largely unavoidable for assessing psychological well-being, self-reported prosocial behaviour is more susceptible to biases such as social desirability and self-perception inaccuracies, which may limit the accuracy of the findings [68]. Prior research has recommended assessing the impact of MBIs on prosocial behaviour using a combination of self-report instruments and behavioural or morally relevant real-world tasks to obtain a more comprehensive evaluation of intervention effects [59]. This recommendation is echoed here, as employing multi-method assessment approaches could better capture prosocial behaviour across micro (individual), meso (interpersonal), and macro (societal) levels [50], thereby strengthening the validity of future findings.

Subsequently, one limitation of the current study is the potential risk of Type I errors due to multiple comparisons, as formal corrections (e.g., Bonferroni or FDR) were not applied to account for this. However, this risk was mitigated by employing a hypothesis-driven approach, focusing on a limited number of pre-specified tests rather than conducting exploratory analyses. Additionally, the use of a linear regression framework to test for moderation effects, followed by a separate ANCOVA sensitivity analysis, adds robustness to the findings. The ANCOVA served as a complementary method to confirm the observed patterns, reducing the likelihood of spurious results. Nonetheless, future research should consider applying formal corrections for multiple comparisons to further minimise the risk of Type I errors.

A further limitation concerns the relatively low internal reliability observed for the Ryff’s Brief Scale of Psychological Well-being in this sample. Reduced scale reliability introduces measurement error, which in turn can weaken observed effect sizes and obscure genuine relationships between variables. Consequently, any lack of significant findings for well-being outcomes should be interpreted with caution, as the limited reliability of the measure may have reduced the study’s sensitivity to detect true intervention effects. Future research should consider using more comprehensive measures of well-being, such as Ryff’s Brief medium or long-form Scale of Psychological Well-being, to enhance reliability and construct validity.

Moreover, as discussed above, because the MBI-CEW combined ethics and wisdom practices, it remains unclear whether the gains reflected the unique contribution of wisdom practices or their interaction with ethics-based training. Future research could therefore incorporate fidelity measures and dismantling designs to clarify the specific and additive effects of distinct meditation components.

## 5. Conclusions

The majority of previous mindfulness research has reported only on group-level effects of interventions, with limited attention to the specific practices participants engage in or how these practices interact with individual differences. The present study helps address this gap by suggesting that individuals high in neuroticism may experience particularly strong prosocial benefits from mindfulness interventions that combine both ethics- and wisdom-based practices. These findings highlight the need to move beyond one-size-fits-all approaches to mindfulness and toward a more nuanced understanding of how distinct meditation practices yield differential effects depending on personality and context. Future research should therefore examine not only which practices are most effective overall, but also for whom and under what conditions they support psychological well-being and prosocial attitudes.

## Figures and Tables

**Figure 1 healthcare-13-03044-f001:**
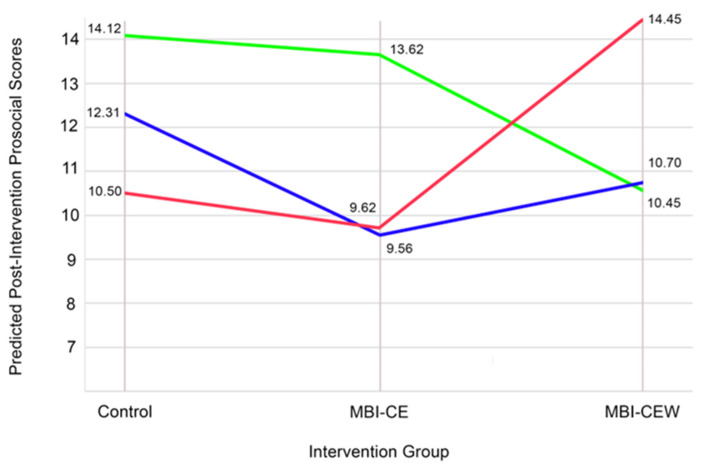
Interaction between intervention group and neuroticism on predicted prosocial scores: Low Neuroticism (−1 SD; green line), Mean Neuroticism (0 SD; blue line), High Neuroticism (+1 SD; red line).

**Figure 2 healthcare-13-03044-f002:**
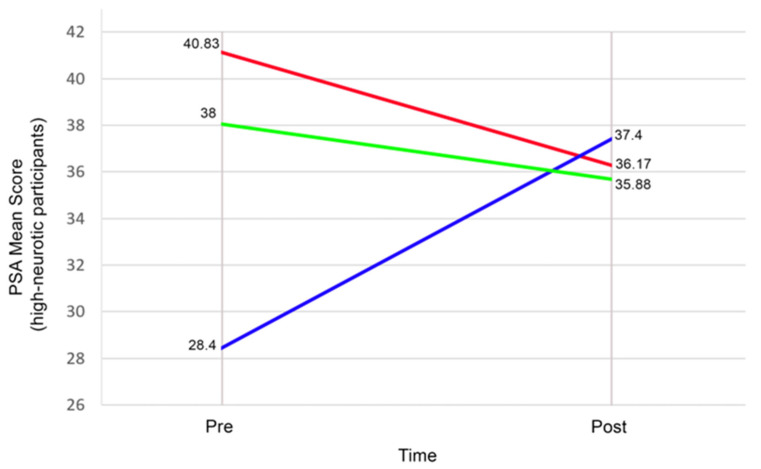
Pre-post changes in mean Prosocial Attitude (PSA) scores among high-neurotic participants for the MBI-CE (red line), MBI-CEW (blue line), and Control (green line).

**Figure 3 healthcare-13-03044-f003:**
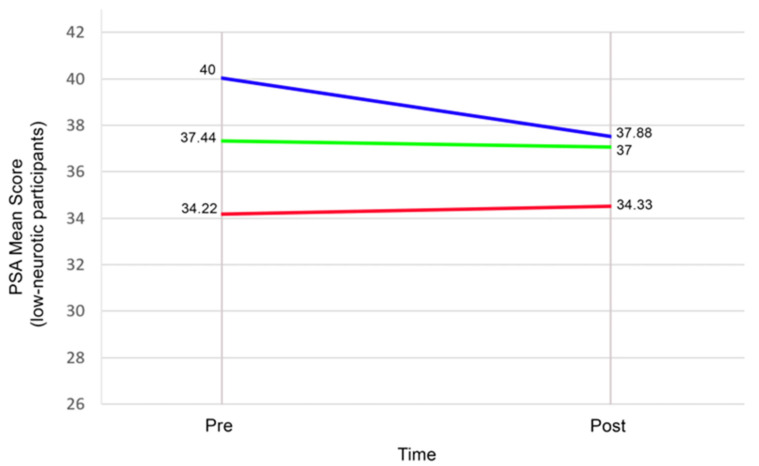
Pre-post changes in mean Prosocial Attitude (PSA) scores among low-neurotic participants for the MBI-CE (red line), MBI-CEW (blue line), and Control (green line).

**Figure 4 healthcare-13-03044-f004:**
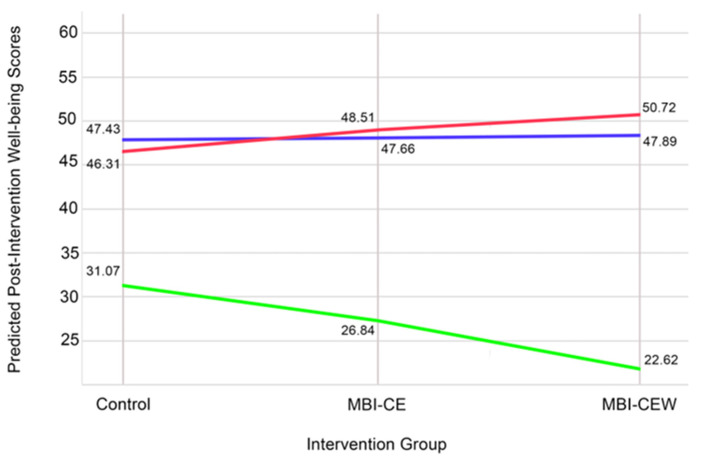
Interaction between intervention group and agreeableness on predicted well-being scores: Low Agreeableness (−1 SD; green line), Mean Agreeableness (0 SD; blue line), High Agreeableness (+1 SD; red line).

**Figure 5 healthcare-13-03044-f005:**
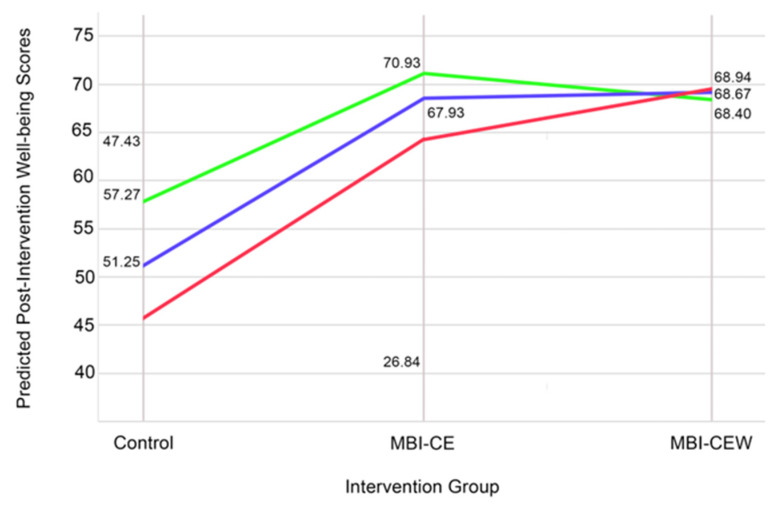
Interaction between intervention group and openness on predicted well-being scores: Low Openness (−1 SD; green line), Mean Openness (0 SD; blue line), High Openness (+1 SD; red line).

**Table 1 healthcare-13-03044-t001:** Means and Standard Deviations of outcome measures pre- and post-intervention by group.

Outcome Measure	Group		Pre-		Post-	
		*n*	Mean	SD	Mean	SD
Prosocialness Scale for Adults (PSA)	MBI-CE	21	38.00	9.04	35.38	10.22
	MBI-CEW	18	33.56	10.35	37.61	6.36
	Control	17	37.71	12.99	36.47	10.70
Ryff’s Brief Scale of Psychological Well-being (PWB)	MBI-CE	21	68.76	6.11	71.29	5.01
	MBI-CEW	18	68.67	6.88	71.78	5.02
	Control	17	36.12	13.04	39.53	12.06

**Table 2 healthcare-13-03044-t002:** Descriptive statistics of trait and outcome variables.

Variable	Mean	SD	Min	Max	Sample Size (N)
Neuroticism (Pre)	7.96	3.29	2	14	56
Agreeableness (Pre)	9.80	2.54	6	14	56
Conscientiousness (Pre)	8.05	2.77	3	14	56
Openness (Pre)	7.70	2.38	2	13	56
Extraversion (Pre)	8.34	2.68	2	14	56
Prosocial (Pre)	36.48	10.77	19	62	56
Prosocial (Post)	36.43	9.20	17	58	56
Well-being (Pre)	58.82	17.51	19	79	56
Well-being (Post)	61.80	16.72	17	88	56

**Table 3 healthcare-13-03044-t003:** Linear regression results for trait on prosocial outcomes.

Statistic	Neuroticism	Agreeableness	Conscientiousness	Openness	Extraversion
Variance Explained (R^2^)	0.578	0.536	0.595	0.549	0.549
F-Statistic (F)	11.17	9.44	12.00	9.92	9.95
*p*-Value (F-Test)	<0.001	<0.001	<0.001	<0.001	<0.001
Baseline Prosocial Scores (β)	0.678 (*p* < 0.001)	0.656 (*p* < 0.001)	0.661 (*p* < 0.001)	0.634, *p* < 0.001	0.653 (*p* < 0.001)
Intervention Group (MBI-CE) (β)	−6.71 (*p* = 0.317)	−1.33 (*p* = 0.545)	10.46 (*p* = 0.101)	−10.41 (*p* = 0.455)	−4.59 (*p* = 0.450)
Intervention Group (MBI-CEW) (β)	−13.44 (*p* = 0.079)	4.03 (*p* = 0.086)	−2.66 (*p* = 0.747)	−5.32 (*p* = 0.674)	3.82 (*p* = 0.524)
Trait (β)	−0.68 (*p* = 0.240)	0.814 (*p* = 0.614)	−0.084 (*p* = 0.896)	−0.28 (*p* = 0.783)	−0.48 (*p* = 0.312)
Interaction (MBI-CE × Trait) (β)	0.70 (*p* = 0.372)	−0.879 (*p* = 0.675)	1.15 (*p* = 0.132)	0.89 (*p* = 0.481)	0.34 (*p* = 0.623)
Interaction (MBI-CEW × Trait) (β)	2.09 (*p* = 0.021)	0.866 (*p* = 0.721)	0.79 (*p* = 0.423)	0.99 (*p* = 0.392)	−0.09 (*p* = 0.897)

**Table 4 healthcare-13-03044-t004:** Linear Regression Results for Trait on Well-being Outcomes.

Statistic	Neuroticism	Agreeableness	Conscientiousness	Openness	Extraversion
Variance Explained (R^2^)	0.852	0.854	0.842	0.854	0.845
F-Statistic (F)	46.98	47.93	43.43	47.74	44.37
*p*-Value (F-Test)	<0.001	<0.001	<0.001	<0.001	<0.001
Baseline Prosocial Scores (β)	0.387 (*p* = 0.001)	0.343 (*p* = 0.002)	0.413 (p<0.001)	0.335 (*p* = 0.003)	0.425 (p=0.001)
Intervention Group (MBI-CE) (β)	7.521 (*p* = 0.343)	6.401 (*p* = 0.403)	28.24 (*p* = 0.001)	4.999 (p=0.720)	7.02 (p<0.001)
Intervention Group (MBI-CEW) (β)	6.751 (*p* = 0.446)	0.879 (*p* = 0.919)	34.31 (*p* = 0.001)	−6.871 (p=0.600)	17.36 (p<0.001)
Trait (β)	−1.648 (*p* = 0.010)	−1.775 (*p* = 0.012)	0.891 (*p* = 0.231)	−2.373 (p=0.026)	−3.29 (p=0.049)
Interaction (MBI-CE × Trait) (β)	1.520 (*p* = 0.068)	1.873 (*p* = 0.044)	−1.260 (*p* = 0.148)	1.192 (p=0.342)	4.56 (p=0.063)
Interaction (MBI-CEW × Trait) (β)	1.671 (*p* = 0.076)	2.701 (*p* = 0.012)	−1.916 (*p* = 0.093)	2.478 (p=0.045)	3.61 (p=0.130)

## Data Availability

The data that support the findings of this study are openly available in the Open Science Framework (OSF) data repository at https://osf.io/cbzp9/?view_only=fe34fee8854f4530a5a04b7ac48ad9d8 (available from 24 April 2024).

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
