# Peer review of "The Moderating Role of Personality on the Effects of Concentration-, Ethics- and Wisdom-Based Meditation Practices for Well-Being and Prosociality"

_healthcare, 2025, doi:10.3390/healthcare13233044_

Round 1
Reviewer 1 Report
Comments and Suggestions for Authors
This carefully designed study contributes to the collective understanding of mindfulness practices in that it seeks to accommodate different personality types, and importantly too, the constructs of compassion, ethics and wisdom as part of the traditional (Buddhist) framings on the practice.
As with most mindfulness studies, an 8 week intervention period is potentially too short to determine effects.
I have wondered too, what was the experience and expertise of the those who facilitated the sessions?
Author Response
Comment 1
Figures and tables can be improved.
Response
Tables:
All tables have been standardized to improve readability and consistency:
- All table text has been resized to 8 pt.
- Vertical column lines have been added to enhance readability across rows.
For Table 1, the names of all outcome measures have been written out in full (no abbreviations) to improve clarity.
Tables 2 and 3 have been moved to Appendix A. We have added a dedicated column indicating the High vs. Low level of each personality trait to make group comparisons immediately clear.
Tables 4 and 5 have been moved to Appendix A. We added: A separate column reporting the number of participants in each cell (for both High and Low levels), and a separate “Planned Contrasts” column that clearly displays the High and Low contrasts and their change values.
Figures
Figures are presented in color to enhance clarity. The MBI-CE is now a solid red line, the MBI-CEW a solid blue line, and the Control a solid green line.
The y-axis titles have been revised to more explicitly describe the outcome depicted. In addition, numeric value markers have been added at the start and end of each line (Pre and Post) to improve readability and facilitate comparison across groups.
Comment 2
I have wondered too, what was the experience and expertise of the those who facilitated the sessions?
Response:
At the end of the intervention section (Line 293) we have included descriptions of the facilitator’s expertise:
“Both the MBI-CE and MBI-CEW were administered by the same facilitator (and primary author) who is trained in MBSR, Meditation Awareness Training, the Buddhist four foundations of mindfulness (Satipaṭṭhāna) practice and Buddhist mindfulness of breathing (Ānāpānasati) practice, as well as holding a NeuroMindfulness coaching certificate accredited by the International Coaching Federation.”
Reviewer 2 Report
Comments and Suggestions for Authors
Please read the attached file.

Author Response
Comment 1
Dichotomization of Continuous Variables
The authors classified personality traits into “high” and “low” groups using a median split.
This approach is widely discouraged in contemporary psychological methodology because it:
- Artificially reduces statistical power,
- Inflates the risk of Type I and Type II errors,
- Introduces arbitrary group distinctions,
- Potentially creates spurious interaction effects.
Recommendation: The authors should acknowledge this limitation clearly and consider re-running moderation analyses using continuous personality variables within linear regression or generalized
We have updated our analytical approach by treating personality traits as continuous variables. Specifically, we employed a linear regression framework to test whether the effect of the intervention on outcomes was moderated by personality traits (see Line 355 for section explaining Statistical Analysis). For significant interactions between intervention group and personality traits, we conducted simple slopes analyses to assess the direction and strength of the relationship between intervention group and outcomes at different levels of the personality trait (e.g., ±1 SD from the mean).
Additionally, to enhance the robustness of our findings, we performed ANCOVA as a sensitivity analysis to provide further evidence regarding the potential moderating role of personality traits, with planned simple-effects contrasts to clarify subgroup patterns (see results section Lines 396 – 720 for detailed changes)
Comment 2
Small and Uneven Cell Sizes
Following subdivision by intervention group and personality category, several cells include very small sample sizes (n < 10), with some subgroup cells even absent (e.g., low Openness in the control group). This leads to:
- Unstable parameter estimates,
- Low statistical power for interaction effects,
- Reduced reliability of effect size calculations.
The discussion should reflect that the results are exploratory rather than confirmatory. Claims of moderation effects—especially those without significant omnibus interaction terms—should be substantially tempered.
This issue has been reduced due to the continuous variable approach (see Line 355 for section explaining Statistical Analysis). However, language has been tempered where appropriate to emphasize the exploratory nature of certain findings within results and discussion sections.
Comment 3
Provide evidence that ANCOVA assumptions were met, or…
Furthermore, in several instances the authors interpret planned contrasts as meaningful despite nonsignificant omnibus Group × Trait interactions. This approach risks overemphasizing isolated or marginal findings.
The primary analysis is now a linear regression framework while the ANCOVA is used as a sensitivity analysis. Nonetheless, consideration of ANCOVA assumptions has been added to the statistical analysis section (Lines 382 – 385).
Language has now been changed to reduce risk of overemphasizing marginal findings, such as “though this finding warrants cautious interpretation” (Line 753) and “this finding should be interpreted with caution, as it was exploratory and not part of the primary analysis.” (Lines 788 – 789)
Comment 4
Theoretical foundation:
To strengthen the theoretical foundation concerning individual differences in mindfulness-related outcomes, it is recommended that the authors integrate the following study:
Saggino, A., Bartoccini, A., Sergi, M. R., Romanelli, R., Macchia, A., & Tommasi, M. (2017). Assessing mindfulness on samples of Italian children and adolescents: The validation of the Italian version of the Child and Adolescent Mindfulness Measure. Mindfulness, 8(5), 1364– 1372. https://doi.org/10.1007/s12671-017-0710-7
We have now added this study to strengthen theoretical grounding: “Although some studies have examined individual differences in mindfulness-related outcomes by gender [63], the influence of personality traits on individual variability in MBI outcomes has not yet been systematically investigated [1].” (Lines 80 – 83)
Thank you for your helpful and constructive comments, which have enabled us to significantly strengthen the paper.
Reviewer 3 Report
Comments and Suggestions for Authors
GENERAL RECOMMENDATION
- Originality (Merit for Publication)
- Strong Originality and Theoretical Contribution: The study is commendable for moving beyond the "one-size-fits-all" approach in mindfulness research and for investigating the moderating role of personality. Specifically, the finding that high neuroticism interacts with wisdom-based practices (MBI-CEW) to enhance prosociality is a novel and significant result.
- Clinical and Applied Relevance: The results have clear implications for clinical and non-clinical practice, suggesting that mindfulness interventions should be tailored to the individual's personality profile.
- Focus on Wisdom: The explicit inclusion and differentiation of the effects of wisdom practices (derived from the Buddhist tradition) is a crucial and often-underestimated area of research that the manuscript effectively addresses.
The manuscript presents a methodologically sound and theoretically relevant study examining an underexplored area in mindfulness research: personality moderation of the effects of specific meditation practices. However, several revisions are necessary before publication.
STRENGTHS
- Theoretical relevance: The focus on individual differences and specific practices represents an important contribution to the mindfulness literature
- Experimental design: The use of three groups (MBI-CE, MBI-CEW, Control) with randomization is appropriate
- Specific focus: The examination of wisdom-based practices as distinct from ethics-based practices is innovative
- Transparency: Data are available on OSF, supporting reproducibility
MAJOR CONCERNS
- Discrepancy between hypotheses and results
Problem: Hypotheses 1-4 predicted that openness, conscientiousness, extraversion, and agreeableness would moderate the effects of both interventions with no differences between MBI-CE and MBI-CEW. However, results show differentiated patterns (e.g., low conscientiousness favors MBI-CEW for prosociality).
Request:
- Revise the hypothesis section to include more specific predictions about how each trait might differentially moderate the two interventions
- In the discussion, explicitly address why the original hypotheses were not confirmed
- Consider whether some hypotheses were too generic given the theoretical framework
- Statistical power and sample size
Problem: With n=56 divided into 3 groups and further stratified by 5 personality traits (high/low), some cells have very small samples (e.g., n=7-8 in many subgroups).
Request:
- The sample size ($N=56$) and subgroup sizes (e.g., High Neuroticism MBI-CEW $n=10$) are very small5555. This raises concerns about statistical power and the generalizability of the findings. The risk of false positives (Type I error) is high, especially given the number of subgroup analyses performed
- Discuss more explicitly how low statistical power limits the interpretability of non-significant results
- Consider whether the median split strategy is optimal or whether dimensional analyses (traits as continuous variables) might be more appropriate
- Multiple testing problem
Problem: Numerous tests were conducted (5 traits × 2 outcomes × 3 group comparisons) without correction for multiple comparisons.
Request:
- Apply a correction (e.g., Bonferroni, FDR) or explicitly justify why it was not used
- At minimum, discuss the risk of Type I errors in the limitations paragraph
- Consider emphasizing more strongly the exploratory approach for non-confirmatory analyses
- MBI-CEW confusion: wisdom vs. wisdom+ethics
Problem: The MBI-CEW group includes both ethics-based practices (Stages 1-2) and wisdom-based practices (Stage 3), making it impossible to isolate the unique contribution of wisdom-based practices.
Request:
- Clarify in the title, abstract, and introduction that MBI-CEW is a cumulative intervention (concentration + ethics + wisdom), not wisdom alone
- Modify conclusions to reflect that benefits may derive from the interaction between ethics and wisdom, not only from wisdom practices
- The authors mention this limitation (lines 609-613), but it should be more prominent
MINOR CONCERNS
- Outcome measures
Problem:
- The PWB scale has modest internal consistency (ω=0.66)
- All measures are self-report, subject to bias
Request:
- Discuss the implications of low PWB reliability (may reduce ability to detect real effects)
- Expand the discussion on self-report limitations (already mentioned, but superficially)
- Consider whether some null findings might be due to measurement issues
- Intervention fidelity control
Problem: There are no manipulation checks to verify whether participants actually practiced/understood the assigned techniques.
Request:
- Although mentioned in limitations (lines 604-612), provide at least descriptive data on:
- Practice adherence (minutes/day from journal entries)
- Adherence differences between groups
- Correlations between adherence and outcomes
- If these data are unavailable, discuss this gap more explicitly
- Interpretation of the neuroticism finding
Problem: The theoretical explanation of the main result (neuroticism × MBI-CEW on prosociality) is very elaborate (lines 497-561), but relies primarily on unmeasured constructs (vulnerable narcissism, interpersonal defensiveness).
Request:
- More clearly distinguish between theoretical speculation and direct empirical evidence
- Consider alternative explanations (e.g., high-neurotic individuals might simply benefit more from variety in practices)
- Cite studies that have actually measured these mediating mechanisms
- Wellbeing analysis
Problem: Effects on wellbeing are generally non-significant or weak, but this is not adequately discussed.
Request:
- Explore why personality moderates prosociality but not wellbeing
- Consider whether wisdom-based practices have more interpersonal than intrapersonal effects
- This could be an important theoretical contribution if adequately developed
- Results presentation
Problem:
- Tables 4-5 are dense and difficult to interpret
- Some key results are buried in the text
Request:
- Simplify tables or add visual annotations (*, **, *** for significance)
- Create a summary figure showing all moderation effects (significant and non-significant)
- Consider moving some descriptive tables (2-3) to supplementary materials
- Minor editorial issues
- Line 1: "Willaim" → "William"
- Lines 40-43: Sentence is too long and convoluted
- Line 96: "mindfulness-based stress reduction (MBSR) program" - provide complete citation (Kabat-Zinn, 1990 or other appropriate)
- Lines 252-256: Inconsistent citation formatting (BCBS, 2025; Salzberg, n.d. - verify if these sources are appropriate for peer-reviewed publication)
- Table 1: "PWR (well-being)" should be "PWB (well-being)"
- Lines 371-375: Figures 1-2 have identical captions except for high/low neurotic - clarify better
SPECIFIC RECOMMENDATIONS
For the Abstract
- Reduce from 43 lines to ~250-300 words maximum
- Make more explicit that MBI-CEW is cumulative (includes CE + W)
- Add main effect size (Hedges' g for the neuroticism effect)
For the Introduction
- Clarify earlier the differences between MBI-CE and MBI-CEW (currently not clear until Methods)
- Reduce the general mindfulness-personality review (lines 47-95) to make room for more theory on why wisdom practices should have differentiated effects
For the Discussion
- Begin by summarizing which hypotheses were supported and which were not
- Give more space to alternative interpretations of the neuroticism finding
- Reduce the length of the theoretical section on vulnerable narcissism (currently ~60 lines) or move part to the introduction
For Limitations
- This section is good but generic. Specify:
- What would be the ideal sample size to detect effects with power 0.80?
- Why weren't behavioral data collected if prosociality was a primary outcome?
- How do specific limitations affect interpretation of null results (not only positive ones)?
QUESTIONS FOR THE AUTHORS
- Have you considered sensitivity analyses using personality traits as continuous variables instead of median split?
- Do data exist on participants' expectations regarding the intervention that might have influenced results?
- Was the facilitator blind to the study hypotheses? If not, could this have influenced intervention delivery?
- Why was follow-up not included if the goal was to examine durable behavioral changes?
- Did control group participants receive any intervention or were they on a waiting list? This could influence motivation and therefore results.
CONCLUSION
This study addresses an important and well-motivated question about personalization of mindfulness interventions. The main finding (neuroticism moderates the effects of wisdom practices on prosociality) is robust and potentially important. However, the manuscript requires substantial revisions to:
- Align hypotheses, methods, results, and interpretation
- Address issues of statistical power and multiple testing
- Clarify what can be concluded from this specific design (wisdom + ethics vs. ethics only)
- Temper theoretical interpretations with greater caution
With these revisions, the manuscript could make a significant contribution to the literature on personalized mindfulness.
Author Response
Comment 1
Problem: Hypotheses 1-4 predicted that openness, conscientiousness, extraversion, and agreeableness would moderate the effects of both interventions with no differences between MBI-CE and MBI-CEW. However, results show differentiated patterns (e.g., low conscientiousness favors MBI-CEW for prosociality).
Request:
- Revise the hypothesis section to include more specific predictions about how each trait might differentially moderate the two interventions
- In the discussion, explicitly address why the original hypotheses were not confirmed
- Consider whether some hypotheses were too generic given the theoretical framework
Hypotheses have now been revised to include more specific predictions how each trait might differentially moderate the two interventions on well-being and prosocial outcomes (see Lines 154 – 248).
The discussion now begins by explicitly addressing why the hypotheses were either confirmed or not (see Lines 745 – 887)
Comment 2
Statistical power and sample size
- The sample size ($N=56$) and subgroup sizes (e.g., High Neuroticism MBI-CEW $n=10$) are very small5555. This raises concerns about statistical power and the generalizability of the findings. The risk of false positives (Type I error) is high, especially given the number of subgroup analyses performed
- Discuss more explicitly how low statistical power limits the interpretability of non-significant results
- Consider whether the median split strategy is optimal or whether dimensional analyses (traits as continuous variables) might be more appropriate
We have updated our analytical approach by treating personality traits as continuous variables. Specifically, we employed a linear regression framework to test whether the effect of the intervention on outcomes was moderated by personality traits (see Line 355 for section explaining Statistical Analysis). For significant interactions between intervention group and personality traits, we conducted simple slopes analyses to assess the direction and strength of the relationship between intervention group and outcomes at different levels of the personality trait (e.g., ±1 SD from the mean).
Additionally, to enhance the robustness of our findings, we performed ANCOVA as a sensitivity analysis to provide further evidence regarding the potential moderating role of personality traits, with planned simple-effects contrasts to clarify subgroup patterns (see results section Lines 396 – 720 for detailed changes)
Comment 3
Multiple testing problem
- Apply a correction (e.g., Bonferroni, FDR) or explicitly justify why it was not used
- At minimum, discuss the risk of Type I errors in the limitations paragraph
- Consider emphasizing more strongly the exploratory approach for non-confirmatory analyses
To address the potential risk of Type I errors, a paragraph has been added to the limitations section (see Lines 956 – 965). This paragraph acknowledges the absence of formal corrections and discusses how the hypothesis-driven approach, combined with the use of a linear regression framework and a separate ANCOVA sensitivity analysis, helps to mitigate the risk of spurious findings. The limitations section also emphasizes the importance of applying formal corrections in future research to further minimize the risk of Type I errors.
Comment 4
MBI-CEW confusion: wisdom vs. wisdom+ethics
Problem: The MBI-CEW group includes both ethics-based practices (Stages 1-2) and wisdom-based practices (Stage 3), making it impossible to isolate the unique contribution of wisdom-based practices.
Request:
- Clarify in the title, abstract, and introduction that MBI-CEW is a cumulative intervention (concentration + ethics + wisdom), not wisdom alone
- Modify conclusions to reflect that benefits may derive from the interaction between ethics and wisdom, not only from wisdom practices
- The authors mention this limitation (lines 609-613), but it should be more prominent
Title, abstract (see Line 25 and Line 39) and introduction (see Lines 133 – 136 and Lines 149 – 152) have been updated to better reflect this. Conclusions have also been modified to better reflect this.
Discussion of the cumulative intervention and what this means for findings has now been added at the end of the discussion section (see Lines 912 – 935).
Comment 5
Outcome measures
- Discuss the implications of low PWB reliability (may reduce ability to detect real effects)
- Expand the discussion on self-report limitations (already mentioned, but superficially)
- Consider whether some null findings might be due to measurement issues
Added a point to the limitations on the implication of low well-being scale reliability (see Lines 966 – 974)
Discussion on self-report limitations has been expanded in the limitations section as well as within the discussion section (see Lines 946 – 955)
Discussion of potential measurement issues has been added in the discussion section (see Lines 845 – 855)
Comment 6
Intervention fidelity control
Problem: There are no manipulation checks to verify whether participants actually practiced/understood the assigned techniques.
Request:
- Although mentioned in limitations (lines 604-612), provide at least descriptive data on:
- Practice adherence (minutes/day from journal entries)
- Adherence differences between groups
- Correlations between adherence and outcomes
- If these data are unavailable, discuss this gap more explicitly
Clarity on this has been added to the intervention design and participants sections:
“Participants were asked to complete at least five entries per week in their daily meditation journals, documenting both formal and informal practice. These journals were submitted to the facilitator as a manipulation check to verify engagement with the assigned techniques. Participants who did not meet this requirement were deemed to have not completed the course and were excluded from the final dataset.” (see Lines 228 – 292)
Comment 7
Interpretation of the neuroticism finding
Problem: The theoretical explanation of the main result (neuroticism × MBI-CEW on prosociality) is very elaborate (lines 497-561), but relies primarily on unmeasured constructs (vulnerable narcissism, interpersonal defensiveness).
Request:
- More clearly distinguish between theoretical speculation and direct empirical evidence
- Consider alternative explanations (e.g., high-neurotic individuals might simply benefit more from variety in practices)
- Cite studies that have actually measured these mediating mechanisms
The section on neuroticism × MBI-CEW on prosociality has been significantly reduced and more clear distinction has been made between theoretical speculations and direct empirical evidence.
Comment 8
Wellbeing analysis
Problem: Effects on wellbeing are generally non-significant or weak, but this is not adequately discussed.
Request:
- Explore why personality moderates prosociality but not wellbeing
- Consider whether wisdom-based practices have more interpersonal than intrapersonal effects
- This could be an important theoretical contribution if adequately developed
More extensive discussion on the well-being findings has now been added to the discussion section explore why, in some cases, personality moderated for prosociality but not wellbeing (see Lines 833 – 855).
- Results presentation
Problem:
- Tables 4-5 are dense and difficult to interpret
- Some key results are buried in the text
Request:
- Simplify tables or add visual annotations (*, **, *** for significance)
- Create a summary figure showing all moderation effects (significant and non-significant)
- Consider moving some descriptive tables (2-3) to supplementary materials
Tables have been redesigned and simplified to improve clarity and readability. Descriptive tables for ANCOVA and planned contrasts have been moved to Appendices.
Tables 2 and 3 have been moved to Appendix A. We have added a dedicated column indicating the High vs. Low level of each personality trait to make group comparisons immediately clear.
Tables 4 and 5 have been moved to Appendix A. We added: A separate column reporting the number of participants in each cell (for both High and Low levels), and a separate “Planned Contrasts” column that clearly displays the High and Low contrasts and their change values.
- Minor editorial issues
- Line 1: "Willaim" → "William"
- Lines 40-43: Sentence is too long and convoluted
- Line 96: "mindfulness-based stress reduction (MBSR) program" - provide complete citation (Kabat-Zinn, 1990 or other appropriate)
- Lines 252-256: Inconsistent citation formatting (BCBS, 2025; Salzberg, n.d. - verify if these sources are appropriate for peer-reviewed publication)
- Table 1: "PWR (well-being)" should be "PWB (well-being)"
- Lines 371-375: Figures 1-2 have identical captions except for high/low neurotic - clarify better
These minor issues have been addressed
SPECIFIC RECOMMENDATIONS
For the Abstract
- Reduce from 43 lines to ~250-300 words maximum
Abstract in within this word limit.
- Make more explicit that MBI-CEW is cumulative (includes CE + W)
This has been made more explicit.
- Add main effect size (Hedges' g for the neuroticism effect)
Effect sizes have now been added.
For the Introduction
- Clarify earlier the differences between MBI-CE and MBI-CEW (currently not clear until Methods)
- Reduce the general mindfulness-personality review (lines 47-95) to make room for more theory on why wisdom practices should have differentiated effects
Title, abstract (see Line 25 and Line 39) and introduction (see Lines 133 – 136 and Lines 149 – 152) have been updated to better reflect this. Conclusions have also been modified to better reflect this.
The mindfulness personality have been slightly reduced and more room have been allowed for exploration of potential differences in practices and on outcomes withing the hypothesis section (see Lines 154 – 249)
For the Discussion
- Begin by summarizing which hypotheses were supported and which were not
- Give more space to alternative interpretations of the neuroticism finding
- Reduce the length of the theoretical section on vulnerable narcissism (currently ~60 lines) or move part to the introduction
Discussion now starts with a summary on hypotheses and the theoretical section on vulnerable narcissism has been reduced.
For Limitations
- This section is good but generic. Specify:
- What would be the ideal sample size to detect effects with power 0.80?
This have now been added (see Lines 941 – 942)
- Why weren't behavioral data collected if prosociality was a primary outcome?
This limitation has now been discussed (see Line 946 – 955)
- How do specific limitations affect interpretation of null results (not only positive ones)?
This limitation has now been discussed (see Line 966 – 974)
QUESTIONS FOR THE AUTHORS
- Have you considered sensitivity analyses using personality traits as continuous variables instead of median split?
We have updated our analytical approach by treating personality traits as continuous variables. Specifically, we employed a linear regression framework to test whether the effect of the intervention on outcomes was moderated by personality traits (see Line 355 for section explaining Statistical Analysis). For significant interactions between intervention group and personality traits, we conducted simple slopes analyses to assess the direction and strength of the relationship between intervention group and outcomes at different levels of the personality trait (e.g., ±1 SD from the mean).
Additionally, to enhance the robustness of our findings, we performed ANCOVA as a sensitivity analysis to provide further evidence regarding the potential moderating role of personality traits, with planned simple-effects contrasts to clarify subgroup patterns (see results section Lines 396 – 720 for detailed changes)
- Do data exist on participants' expectations regarding the intervention that might have influenced results?
This data was not collected however see answer below for explanation of evaluation checks of both instructor and the intervention.
- Was the facilitator blind to the study hypotheses? If not, could this have influenced intervention delivery?
The facilitator was not blind to the study hypotheses. However, to address this, Intervention Delivery Checks were implemented at the end of the course, where participants evaluated both the instructor and the intervention. These evaluations have now been detailed in the measures and findings sections of the manuscript to ensure transparency and to account for any potential bias in intervention delivery. (see Lines 334 – 344 and Lines 722 – 731)
- Why was follow-up not included if the goal was to examine durable behavioral changes?
We fully acknowledge the importance of including a follow-up assessment to examine the durability of behavioural changes. Unfortunately, this was not feasible within the present study due to timing constraints: data collection occurred during the final weeks of the university semester, and participants were subsequently unavailable for further contact once the term ended. This limitation has been explicitly noted in the manuscript (see Lines 941 – 943).
- Did control group participants receive any intervention or were they on a waiting list? This could influence motivation and therefore results.
They were a waiting list control. This is now made clear on Line 28 and Line 263
Thank you for your helpful and constructive comments, which have enabled us to significantly strengthen the paper.
Round 2
Reviewer 2 Report
Comments and Suggestions for Authors
Accept in this form